# Structure of *Mycobacterium tuberculosis* Cya, an evolutionary ancestor of the mammalian membrane adenylyl cyclases

Ved Mehta[1†], Basavraj Khanppnavar[1,2†], Dina Schuster[1,2,3], Ilayda Kantarci[1], Irene Vercellino[1], Angela Kosturanova[1], Tarun Iype[1], Sasa Stefanic[4], Paola Picotti[3], Volodymyr M Korkhov[1,2]*

[1]Laboratory of Biomolecular Research, Division of Biology and Chemistry, Paul Scherrer Institute, Villigen, Switzerland; [2]Institute of Molecular Biology and Biophysics, ETH Zurich, Zurich, Switzerland; [3]Institute of Molecular Systems Biology, ETH Zurich, Zurich, Switzerland; [4]Institute of Parasitology, University of Zurich, Zurich, Switzerland

*For correspondence:
volodymyr.korkhov@psi.ch

†These authors contributed equally to this work

Competing interest: The authors declare that no competing interests exist.

**Abstract** *Mycobacterium tuberculosis* adenylyl cyclase (AC) Rv1625c/Cya is an evolutionary ancestor of the mammalian membrane ACs and a model system for studies of their structure and function. Although the vital role of ACs in cellular signalling is well established, the function of their transmembrane (TM) regions remains unknown. Here, we describe the cryo-EM structure of Cya bound to a stabilizing nanobody at 3.6 Å resolution. The TM helices 1–5 form a structurally conserved domain that facilitates the assembly of the helical and catalytic domains. The TM region contains discrete pockets accessible from the extracellular and cytosolic side of the membrane. Neutralization of the negatively charged extracellular pocket Ex1 destabilizes the cytosolic helical domain and reduces the catalytic activity of the enzyme. The TM domain acts as a functional component of Cya, guiding the assembly of the catalytic domain and providing the means for direct regulation of catalytic activity in response to extracellular ligands.

## Editor's evaluation

This manuscript reports the first full-length structure of membrane-bound adenylyl cyclase from the pathogen *Mycobacterium tuberculosis*. The structure provides insights into its potential mechanism of action and reveals similarities to its mammalian counterpart. Thus, this paper is of potential interest to a broad audience including the fields of infectious diseases, signaling, and evolutionary biologists.

## Introduction

Adenylyl cyclases (ACs) convert molecules of ATP into 3,5-cyclic AMP (cAMP), a universal second messenger and a master regulator of cellular homeostasis (*Linder, 2006*). In mammalian cells, the membrane-associated ACs (*Figure 1A*) generate cAMP upon activation of the cell surface receptors, GPCR, via G protein subunits (*Sassone-Corsi, 2012*), or in some cases by $Ca^{2+}$/calmodulin (*Halls and Cooper, 2011*). The cAMP molecules produced by the ACs bind to a number of effector proteins, including protein kinase A (*Halls and Cooper, 2017*), cyclic nucleotide-gated channel channels (*Kaupp and Seifert, 2002*), exchange protein activated by cAMP (EPAC) (*de Rooij et al., 1998*), popeye proteins (*Schindler and Brand, 2016*), among others, which in turn regulate virtually all aspects of cellular physiology (*Yan et al., 2016*). The nine mammalian membrane ACs (AC1–9)

**Figure 1.** Structure of Cya–NB4 complex. (**A**) Schematic representation of the mammalian membrane adenylyl cyclases (ACs), indicating the key elements of AC structure: 12 transmembrane (TM) domains, 2 catalytic domains, an ATP, and a forskolin (Fsk)-binding site. The protein is depicted in a G-protein-bound state. (**B, C**) A schematic representation of Rv1625c/Cya, illustrating the regions resolved in the cryo-EM structure. The TM region is coloured orange, the helical domain (HD) is green, the catalytic domain is blue. Regions absent in the cryo-EM structure are grey. (**D**) The activity of the full-length Cya in detergent is similar in the absence (yellow) and in the presence of nanobody NB4 (pink); the soluble domain of Cya (SOL, blue) shows low levels of activity. The activity of SOL in the presence of NB4 (cyan) is similar to SOL alone. For all experiments, the data are shown as mean ± standard error of the mean (SEM) (n = 3; for SOL, n = 6). (**E**) The density map of Cya–NB4 complex at 3.57 Å resolution, obtained using masked refinement of the best dataset with C2 symmetry imposed. (**F**) The corresponding views of the atomic model of Cya–NB4 complex, coloured as in B, C. 'N' indicates the N-terminal part of the protein; 'HD' – helical domain; 'CAT' – catalytic domain.

The online version of this article includes the following source data and figure supplement(s) for figure 1:

**Source data 1.** Adenylyl cyclase activity data and analysis (*Figure 1D*).

**Figure supplement 1.** Purification and characterization of Cya.

**Figure supplement 2.** Cryo-EM processing workflow.

**Figure supplement 3.** Properties of the Cya–NB4 density maps.

**Figure supplement 4.** Cryo-EM density map and model of Cya in C1 symmetry.

**Figure supplement 5.** X-ray structure of the catalytic domain of Cya, Cya-SOL, bound to NB4.

share the topology and domain organization: 12 transmembrane (TM) helices with TM6 and TM12 extending to form a coiled coil of the helical domain (HD), linking the TM bundle to the bipartite catalytic domain (*Figure 1A*; *Qi et al., 2019*). Recently, we determined the cryo-EM structure of the full-length AC, the bovine AC9 bound to G protein $\alpha_s$ subunit, revealing the organization of the

membrane-integral region of a membrane AC (*Qi et al., 2019*). Although the structure provided important insights into the mechanism of AC9 auto-regulation, the role of the elaborate 12-helical membrane domain remains unexplained.

A putative evolutionary ancestor of the mammalian membrane ACs has been identified in the genome of *Mycobacterium tuberculosis*: Rv1625c, or Cya (*Guo et al., 2001*; *Guo et al., 2005*); for simplicity, we refer to this protein as Cya throughout. This protein is one of the sixteen ACs present in genome of *M. tuberculosis*, and one of five ACs predicted to be polytopic membrane proteins (*Bai et al., 2011*). The exact function of Cya is not clear, although available evidence indicates that the protein may be involved in $CO_2$ sensing (*Townsend et al., 2009*) and cholesterol utilization by *M. tuberculosis* (*Johnson et al., 2017*; *VanderVen et al., 2015*). Cholesterol utilization during infection by *M. tuberculosis* is linked to its pathogenesis (*Wilburn et al., 2018*), indicating a potential role of Cya at some stages of macrophage infection. The catalytic domain of Cya belongs to the class III AC/guanylyl cyclase (GC) fold, similar to the mammalian membrane ACs. Unlike the mammalian ACs, Cya is predicted to include only six TM helices, with TM6 extending into a HD connected to the catalytic domain (*Figure 1B, C*). The protein has to dimerize to form a functional unit that has been previously presumed to resemble the pseudo-heterodimeric fold of the full-length mammalian ACs (*Ketkar et al., 2006*; *Vercellino et al., 2017*). Although the structures of the *M. tuberculosis* Cya-soluble domain in monomeric form (*Ketkar et al., 2006*) and that of the homologous *M. intracellulare* Cya in dimeric form *Vercellino et al., 2017* have been solved using X-ray crystallography, a structure of a full-length mycobacterial AC that includes the TM region has not been determined until now.

The role of the membrane domain in membrane ACs is a mystery. Polytopic membrane nucleotidyl cyclases with membrane domains of known function have been described in several organisms. The ACs in *Paramecium*, *Plasmodium*, and *Tetrahymena* are fused to an ion channel module (*Weber et al., 2004*), and the light-sensitive GCs in several fungi, such as *Blastocladiella emersonii*, which are fused to a rhodopsin-like membrane module that binds a light-sensitive retinal chromophore (*Avelar et al., 2014*). However, the functional role of the TM regions in the mammalian membrane ACs remains unclear. Although our structure of the bovine AC9 provided a description of the unique TM helix arrangement in the membrane anchor of the protein, it shed relatively little light on the possible function of the membrane domain (*Qi et al., 2019*). Interestingly, experiments with domain-substituted Cya, a presumptive evolutionary ancestor of the mammalian ACs, have suggested that its membrane region may have a regulatory role, potentially acting as a receptor for yet unidentified ligands (*Beltz et al., 2016*).

Understanding the structure and function of the AC membrane domains, conserved through evolution from bacteria to mammals, is essential for understanding the regulation of cAMP generation by the cells at rest and during AC activation. The importance and necessity of a complex polytopic membrane domain in the membrane ACs is one of the key open questions in the cAMP signalling field. To address this key question, we set out to determine the structure of the model membrane AC, *M. tuberculosis* Cya.

## Results
### Characterization of the full-length Cya

The full-length *M. tuberculosis* Cya (*Figure 1C*) was expressed in *Escherichia coli* and purified in digitonin micelles using affinity chromatography followed by size-exclusion chromatography (SEC; *Figure 1—figure supplement 1*). AC activity assays confirmed that the full-length protein was purified in a functional form (*Figure 1D*; $K_m$ for Cya was ~80 μM). The 'SOL' construct, consisting of the catalytic cytosolic domain (residues 203–428) showed low activity (*Figure 1D*), indicating the importance of the membrane-spanning region for proper assembly and activity of the cyclase, in agreement with the previous reports (*Vercellino et al., 2017*; *Ding et al., 2005*). Previously reported SOL activity values, obtained under different experimental conditions in the presence of low salt and 22% glycerol, showed concentration dependence of the enzymatic activity, consistent with the requirement of the SOL domain to dimerize to form a functionally active unit (*Guo et al., 2001*).

## Nanobody NB4 facilitates Cya structure determination

Cya is a relatively small membrane protein (45 kDa for a monomer). The presence of an HD linking the TM domain with the catalytic domain of Cya makes this protein a challenging target for structural studies. To increase the likelihood of high-resolution structure determination, we used the purified Cya to generate a panel of nanobodies, camelid antibody fragments (*Geertsma and Dutzler, 2011*), recognizing the target protein with high affinity. One of these reagents, nanobody 4 (NB4), had no effect on the catalytic activity of the full-length cyclase (*Figure 1D*), but had a nanomolar affinity for the SOL domain (*Figure 1—figure supplement 1E*). We reconstituted a complex of the detergent-purified full-length Cya and NB4 (mixed at a molar ratio of 1:1.5), in the presence of 0.5 mM MANT-GTP (a non-cyclizable nucleotide-derived AC inhibitor), and 5 mM $MnCl_2$. The sample was subjected to cryo-EM imaging and single particle analysis (*Figure 1—figure supplement 2*), yielding a 3D reconstruction of the protein in C1 symmetry at 3.8 Å resolution (*Figure 1—figure supplements 2–4*, *Supplementary file 1*).

The reconstruction revealed the full-length Cya arranged as a dimer bound to three copies of NB4 nanobody: two copies bound symmetrically to the SOL portions of the Cya dimer, and one asymmetrically bound to the extracellular surface of the protein (*Figure 1—figure supplement 3*). To visualize the details of the Cya–NB4 interaction, we crystallized the SOL construct in the presence of NB4 and solved the X-ray structure of the complex at 2.1 Å resolution (*Figure 1—figure supplement 5A*, *Supplementary file 2*). The structure showed an extensive interaction interface between the negatively charged surface of the monomeric SOL domain and the NB4 (*Figure 1—figure supplement 5B*). Interestingly, the crystallized construct did not form a native-like dimeric form of the enzyme, but nevertheless retained the ability to bind to MANT-GTP/$Mn^{2+}$, with an unusual twist of the MANT-GTP base (*Figure 1—figure supplement 5C*). The well-resolved structure of the Cya-SOL–NB4 complex allowed us to reliably place NB4 into the cryo-EM density map (*Figure 1—figure supplements 3 and 4*).

To improve the resolution of the cryo-EM density map in the regions of highest interest, we masked out the extracellular NB4 density and refined the Cya–NB4 dataset imposing the C2 symmetry. This resulted in a 3D reconstruction at 3.57 Å resolution, which allowed us to reliably trace the polypeptide chain in the cryo-EM density map, covering residues 41–428 of the full-length Cya construct (*Figure 1E, F*, *Figure 1—figure supplement 3*, *Supplementary file 1*).

## Key features of the Cya structure

Our 3D reconstruction revealed the previously unresolved portion of the protein, the 6-TM bundle, arranged into a homodimer (*Figures 1F and 2B, D*). The SOL portion of the protein, linked to the TM region via the HD, adopted a conformation consistent with our previous structure of the SOL domain of *M. intracellulare* Cya homologue (*Figure 1—figure supplement 5*; *Vercellino et al., 2017*). The two nucleotide-binding sites of Cya are occupied with the molecules of MANT-GTP/$Mn^{2+}$, which we modelled based on the previous structures and the X-ray structure of SOL–NB4 complex (*Figure 1—figure supplement 5*). The C1 and C2 maps provide no clear evidence of asymmetry in the active site, which we have observed in the structure of *M. intracellulare* Cya (*Vercellino et al., 2017*). Therefore, the two MANT-GTP molecules were modelled in identical orientations.

The HD region is believed to be a critical element in the membrane and soluble ACs and GCs, as this region couples the N-terminal regulatory domains to the catalytic function of these proteins (*Kang et al., 2019*; *Scheib et al., 2018*; *Ohki et al., 2017*). In Cya, the HD extends from the TM6 (*Figure 2A, B*), forming a coiled coil observed in the structures of homologous proteins, including AC9 (*Figure 2C*; *Qi et al., 2019*) and sGC (*Kang et al., 2019*; *Figure 3*). Interestingly, the size difference between the HD helix in Cya and the HD1 and HD2 helices in AC9 leads to an ~90 rotation of the corresponding TM regions, relative to the catalytic domains (*Figure 2—figure supplement 1A*). This may be an indication that the exact structural alignment of the TM domain and the relatively remote catalytic domain may not be a conserved feature of the membrane ACs. Instead, it is likely that the precise TM–HD and HD–catalytic domain coupling plays the key role in the formation and regulation of the catalytic centre in the membrane AC, consistent with the function of the HD as a transducer element in the AC structure (*Ziegler et al., 2017*).

The resolved portion of the Cya N-terminus (residues $V_{41}ARRQR_{46}$), rich in positively charged residues, is immediately adjacent to the HD region. The early work on Cya identified the mutations in this

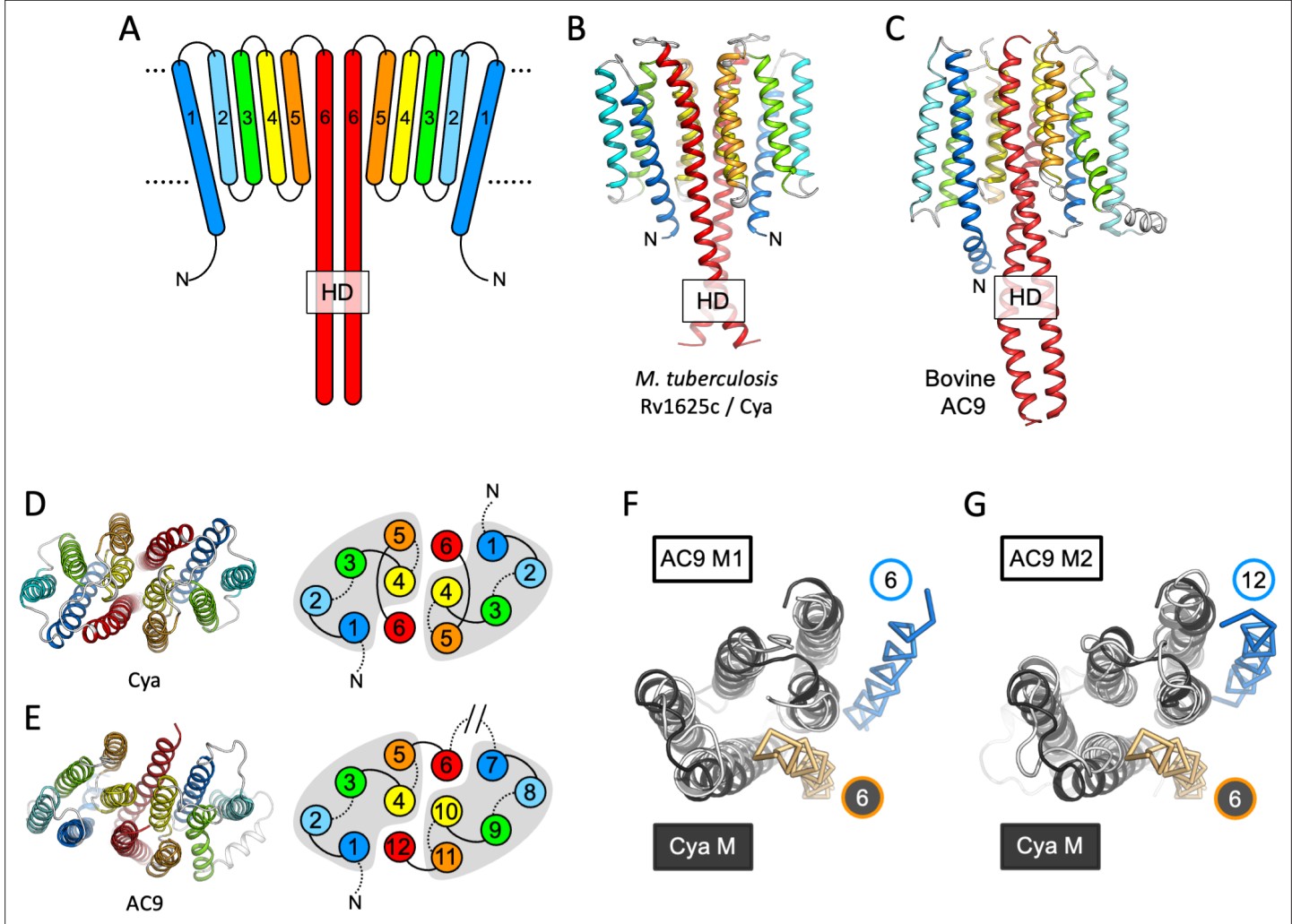

**Figure 2.** Features of the Cya transmembrane (TM) domain. (**A**) A schematic representation of the 6-TM bundles (TM1–TM6) of Cya, arranged as dimer. A view of Cya (**B**) and AC9 (**C**) TM domain parallel to the membrane plane. A view of Cya (**D**) and AC9 (**E**) TM domain perpendicular to the membrane plane. The schematic indicates the relative arrangement of the TM helices, with helices 4, 5, and 6 at the dimer interface. The grey shapes indicate the conserved structural motif (TM1–5 in Cya, TM1–5 and TM7–11 in AC9) of the membrane adenylyl cyclases (ACs). The extracellular and intracellular loops connecting the TM helices are shown using solid and dotted lines, respectively. The connection between TM6 and TM7 of AC9 is indicated as a broken line, in place of the catalytic domain C1a and the connecting loop C1b. Alignment of the 6-TM bundles of Cya (black) and AC9 (white) reveals a high level of structural conservation, in particular in the TM1–5 ( (RMSD) 3.42 Å over 112 residues; **F**) and TM7–11 regions of the two proteins (RMSD 3.56 over 112 residues; **G**). The positions of the helical domain (HD)-forming TM helices are conserved, but the TM6 and TM12 are swapped in AC9 (blue) relative to Cya (orange).

The online version of this article includes the following figure supplement(s) for figure 2:

**Figure supplement 1.** Comparison of *M. tuberculosis* Cya and bovine AC9.

**Figure supplement 2.** Sequence conservation of the transmembrane (TM) domain residues of mycobacterial Cya homologues.

**Figure supplement 3.** Conservation of the extracellular pockets of Cya.

region that disrupt the function of the protein (*Vercellino et al., 2017*), suggesting that the intact residues in the N-terminus stabilize the HD. Our structure provides the structural basis for understanding the likely disruptive effects of these mutations. The positively charged residues R43–R44 likely stabilize the negatively charged surface of the HD (*Figure 2—figure supplement 1B*).

## The TM1–5 bundle as a rail for the HD helices

The TM helices 4, 5, and 6 of Cya form an extensive dimer interface within the membrane (*Figure 2D*). The dimer interface residues, close to the 'core' of the protein, are relatively well conserved among

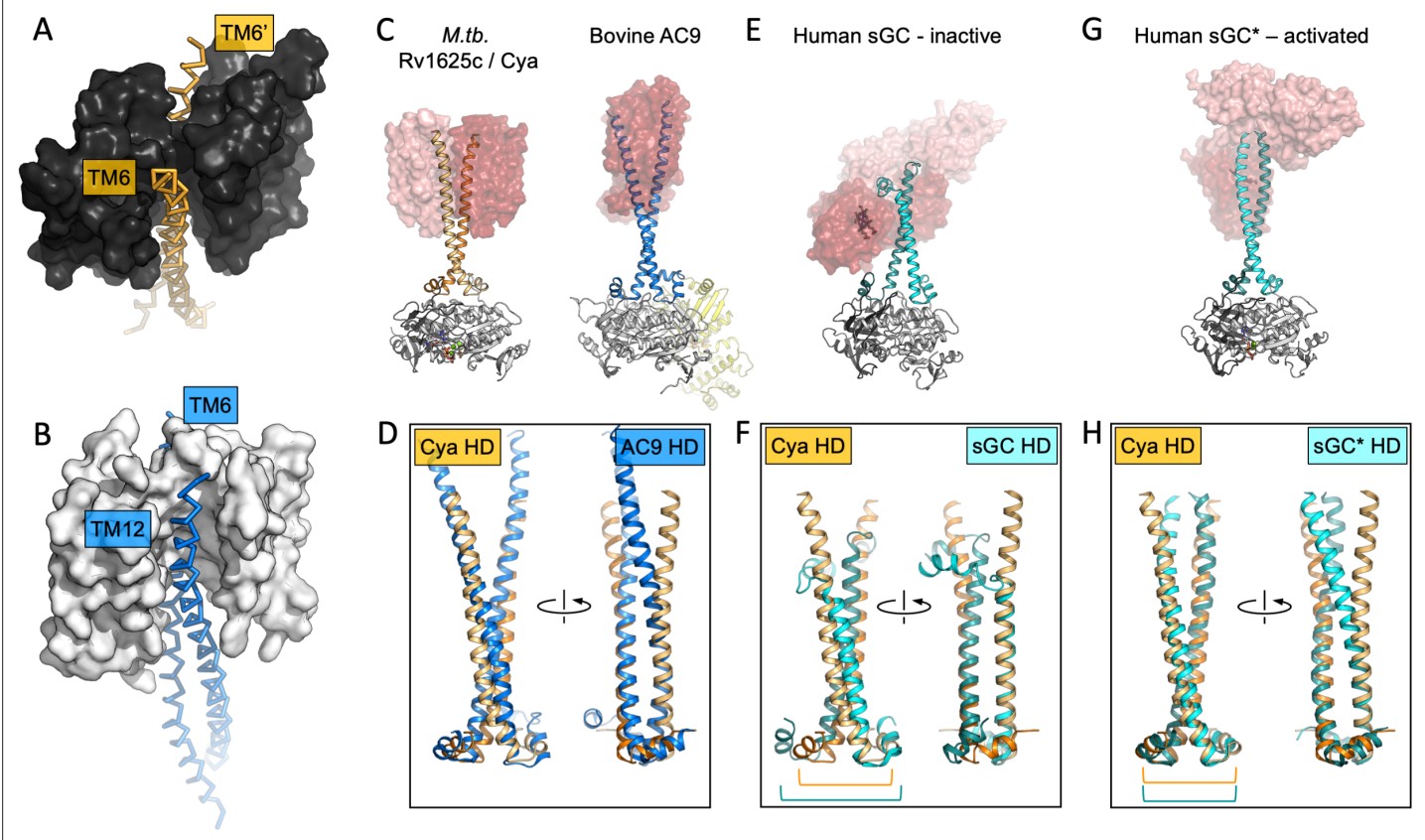

**Figure 3.** The role of the transmembrane (TM) domain as a chaperone for helical domain (HD)/adenylyl cyclase (AC) assembly. Guide rail-like structures are formed by the TM1–5 in Cya (black, **A**) and the TM1–5/TM7–11 in AC9 (white, **B**). The arrangement of these rail-like structures positions the TM6 helices for optimal assembly of the HD and the catalytic domain. (**C**) The views of Cya and AC9 with the TM1–5 (Cya) and TM1–5/TM7–11 (AC9; PDB ID: 6r3q) represented as transparent surface. (**D**) The TM–HD regions of Cya and AC9 aligned. Despite the difference in HD length and the deviation in the TM domains, the cores of the HD domains are well aligned. (**E, F**) Similar to C, D, for the human soluble guanylyl cyclase sGC (inactive form; PDB ID: 6jt0). (**G, H**) Similar to E, F, for the activated form of sGC (sGC*; PDB ID: 6jt2). The brackets in F and H indicate the misaligned (**F**) and aligned (**H**) portions of the Cya and sGC HDs.

the Cya homologues from *Mycobacteria* (*Figure 2—figure supplements 2 and 3*), with relatively poorly conserved residues in TM1–3. A comparison of the 6-TM bundle of Cya with the corresponding regions in the bovine AC9 (TM1–6 and TM7–12) shows that the helices TM1–5 (and TM7–11 for the AC9) form a well defined structural motif (*Figure 2D, E*). A striking difference between the Cya and AC9 membrane domains is that the TM region that forms the HD helix is swapped in AC9: the TM12 of AC9 occupies the same position as the TM6 of Cya. Similarly, TM6 in AC9 is placed in a corresponding position relative to the TM7–11 (*Figure 2F, G*). The TM1–5 bundle in Cya appears to act as a 'guide rail' for the TM6/HD helix of Cya, guiding the correct assembly of the HD coiled coil and the catalytic domain of the cyclase (*Figure 3A*). This feature is remarkably similar in AC9, with TM1–5 and TM7–11 arranged in a near-identical way (*Figure 3B*), and with a closely matching HD core (*Figure 3D*).

The previous experiments in *M. intracellulare* Cya have shown that the HD and the TM regions of the protein are critically important for the protein's dimerization and functional assembly (*Vercellino et al., 2017*). The lack of the TM region results in failure to form a stable active dimer of *M. tuberculosis* Rv1625c/Cya, even in the presence of a nucleotide analogue MANT-GTP, judged by the inability of MANT-GTP to induce crystallization of the protein in a dimeric form (*Figure 1—figure supplement 5*). In contrast, the soluble domain of the *M. intracellulare* Cya is effectively dimerized by MANT-GTP (*Vercellino et al., 2017*). The importance of the TM domain as a factor that promotes correct protein folding is further illustrated by the ability of the isolated Cya-SOL construct to form an inactive domain-swapped dimeric assembly (*Barathy et al., 2014*). It is thus tempting to suggest

that the key function of the TM domain in a membrane AC is to guide the assembly of the enzyme in a catalytically competent form.

This may have important implications for AC regulation. In a related enzyme, the NO-sensing sGC, the heme-containing NO-receptor domain is fused to the HD region in place of the TM regions seen in Cya or in the mammalian AC9 (*Figure 3E–G*). In its inactive form, the sGC displays a conformation where HD helices are bent, with an accompanying substantial unwinding of the HD core (*Figure 3E*). Comparison of the Cya HD core with that of the sGC HD core highlights this discrepancy (*Figure 3F*). In contrast, activation of sGC is accompanied with a large-scale conformational change, 'straightening' the HD (*Figure 3H*) and adopting the HD conformation that closely matches that of Cya (*Figure 3H*). The position of the 'kink' in the HD of sGC approximately corresponds to the membrane–cytosol interface in Cya. Thus, the very distant yet related proteins sGC and Cya (as well as AC9 and other membrane ACs) may be subject to very similar modes of regulation involving changes in the HD, which may result in changes in the catalytic domain of the protein. While in sGC the process is guided by the heme-containing receptor domain, in the membrane ACs this function is likely performed by the TM domain.

## The TM domain of Cya as a putative receptor module

The structure of Cya revealed several prominent cavities in the TM domain of the protein, which may serve a stabilizing or regulatory role (*Figure 4*). A negatively charged cleft (site Ex1) is formed at the extracellular interface of the two 6-TM bundles (*Figure 4A, D, E*). The negative charge of this pocket is provided by the residues D123, E164, and D170 of each monomer, facing into the cavity (*Figure 4E*). This region may be involved in binding of positively charged ions, small molecules, lipids, or peptides. The ability of NB4 nanobody to interact with this pocket spuriously (*Figure 1—figure supplement 4*, *Figure 4—figure supplement 1*) indicates that it may also be a site of interaction with a yet unknown natural protein partner. Additionally, a prominent pocket open to the extracellular side of the protein is formed within each TM bundle (site Ex2; *Figure 4A, C, D*). This pocket may accommodate small molecules or lipids, with a possible access route from the outer leaflet of the lipid bilayer surrounding Cya. A similar internal pocket is present in the TM1–6 bundle in AC9 (*Figure 4—figure supplement 2*).

Deep pockets on the cytosolic side of the Cya TM region are formed between the HD domain and the N-terminus/TM1 (site Cy1), as well as between TM1 and TM3 (site Cy2; *Figure 4B–D*). Close to the entrances into these pockets are the positively charged residues R43 (Cy1/Cy2), R44 (Cy2), R46 (Cy1) of the N-terminus, as well as R203 and R207 (Cy1) in the HD domain from the adjacent monomer (*Figure 4D*). The positive charge of this region indicates a potential role in interactions with the phospholipid headgroups or positively charged peptides or small molecules. The interpretation of these cytosolic intramembrane pockets requires caution, as the residues 1–40 of Cya are not resolved in our 3D reconstruction but may interact with and occlude these pockets. Analysis of the area around the sites Ex2–Cy2 shows that they are discontinuous, precluding formation of a channel traversing the entire width of the membrane (*Figure 4D*). Our molecular dynamics (MD) simulations confirmed the ability of water molecules to enter into the Ex2 and Cy2 site, but no TM water transport could be observed (*Figure 4—figure supplement 3*). Thus, while the pockets Ex2 and Cy2 provide support to the hypothesis of the AC TM domain as a receptor, any translocation events would have to involve substantial conformational rearrangements opening the connection between the two sites.

Our density map features a small but prominent density in the site Ex1 (*Figure 4E*, *Figure 4—figure supplement 1*), which presently cannot be assigned to a specific entity, but which could correspond to a bound metal (Na$^+$, Mg$^2$, Mn$^2$, or a yet unknown component co-purified with the protein from *E. coli*). It is conceivable that disruption of this negatively charged interface may lead to the loss of the rail structure of the TM region, with concomitant changes in the HD helix arrangement and ultimately the catalytic domain of Cya. To test the behaviour of this site, we performed MD simulations (*Figure 4—figure supplement 3*). The site Ex1 behaved as a genuine metal-binding site, occupied by the K$^+$ and Mg$^{2+}$ ions similar to the metal-binding site in the Cya catalytic centre over the course of the simulation (*Figure 4—figure supplement 3B–E*). Thus, the evidence obtained experimentally and using MD simulations strongly supports a function of the extracellular surface of Cya as a receptor for positively charged ligands.

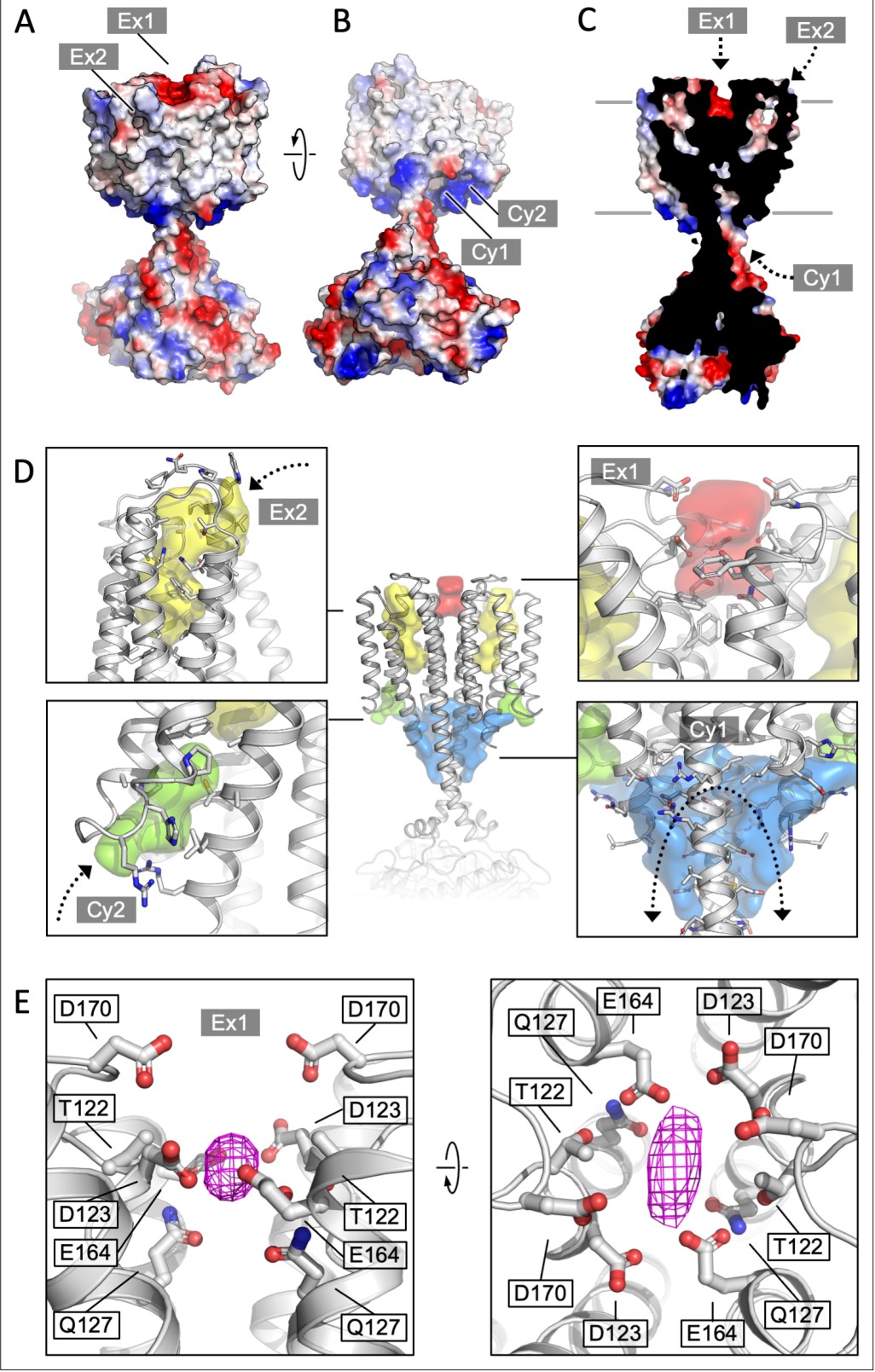

**Figure 4.** Cya transmembrane (TM) domain as a receptor module. (**A**) A view of the Cya structure (nanobody NB4 not shown) in surface representation, coloured according to electrostatic potential. The location of two putative binding sites, sites Ex1 (negatively charge) and Ex2 (hydrophobic) are indicated. (**B**) Similar as (**A**), a view at the Cya structure from the cytosol, showing the locations of the positively charged sites Cy1 and Cy2. (**C**) A slice through

*Figure 4 continued on next page*

*Figure 4 continued*

the structure shows the internal cavities with access points Ex1, Ex2, and Cy1. (**D**) The density maps corresponding to the internal cavities within the TM region of Cya, calculated using 3 V (**Voss and Gerstein, 2010**) and low pass filtered to 3 Å for presentation purposes. Arrows indicate the access to the cavities. (**E**) A prominent density featured in the density map of Cya–NB4 complex, occupying the site Ex1. Polar and negatively charged residues surround the density, consistent with a binding site for metals (or organic cations).

The online version of this article includes the following figure supplement(s) for figure 4:

**Figure supplement 1.** Density present in the Ex1 site.

**Figure supplement 2.** Comparison of the intramembrane cavities formed by Cya and AC9.

**Figure supplement 3.** Molecular dynamics (MD) simulations of Cya.

## The Ex1 site controls the helical and catalytic domain

To test the role of the Ex1 site experimentally, we mutated the polar residues lining this site (T122, D123, Q127, E164, and D170) to Ala (**Figure 5A–C**). The resulting construct (referred to as Ex1-5A, **Figure 5A, C**) was successfully expressed and purified in digitonin (**Figure 5—figure supplement 1A**). The thermostability profiles of the Ex1-5A mutant in the absence or in the presence of a nucleotide were similar to those of the wild-type Cya (**Figure 5D**, **Figure 5—figure supplement 1B, C**). In contrast, the enzymatic activity of Ex1-5A was substantially reduced (with a dramatic increase in apparent $K_m$, **Figure 5E**), with a fourfold reduction in apparent affinity for a nucleotide inhibitor, MANT-GTP (judged by the MANT-GTP $IC_{50}$ values, **Figure 5F**).

To investigate the effects of the Ex1-5A mutant on Cya structure, we performed limited proteolysis-coupled mass spectrometry (LiP-MS) experiments on both the wild type and the mutant in the absence of added nucleotides, and compared the peptides obtained by pulse proteolysis with proteinase K (PK) (**Figure 5G**). Comparative analysis of the LiP-MS profile of the wild-type Cya and the Ex1-5A mutant revealed a significant increase in protease accessibility of the HD in the mutant (**Figure 5G, H**). Thus, modification of the extracellular site Ex1 of Cya leads to changes in the dynamics of its cytosolic HD, accompanied by a dramatic reduction in enzymatic activity.

Interestingly, only two of the site Ex1 residues are well conserved, based on the alignment of 180 close homologues of Cya (**Figure 2—figure supplements 2 and 3A**): while T122 and Q127 are relatively well conserved, the residues D123, E164, and D170 are not. Although our evidence points to a possibility of *M. tuberculosis* Cya Ex1 site's involvement in cation binding, this region may play distinct roles in the Cya homologues of other mycobacterial species. Likewise, the site Ex2 is lined by largely nonconserved residues (**Figure 2—figure supplement 3B**). Developing a more precise understanding of this site's functional role will be a prerequisite for understanding the significance of the few conserved residues in this pocket (F87, A109, G125 in the *M. tuberculosis* Cya).

To further support our insights derived from the experiments with the Ex1-5A mutant we performed the MD simulations with Ex1-5A protein (**Figure 5—figure supplement 2**), using the identical protocol as those used for the wild-type Cya (**Figure 4—figure supplement 3**). The simulation revealed a dramatic difference between the *B* factors of the Ex1-5A and the wild-type protein, indicative of increased conformational flexibility of the mutant (**Figure 5—figure supplement 2A**). As expected, metal ion binding was preserved at the cytosolic domain of the Ex1-5A, but was completely disrupted at the extracellular Ex1 site (**Figure 5—figure supplement 2B, C**). Analysis of the distances between select residues in Cya and in Ex1-5A showed that although the core remains relatively stable for both protein, the TM1–5 region may be more conformationally flexible (judged by the increased average distances between the residues A77–A77 and L92–L92 within the dimer; **Figure 5—figure supplement 2D, E**). The same is true for portions of the catalytic domain, evident from the increased average distance in the residue Q289. Together with the functional data, the results of the MD simulations strongly suggest that disruption of the extracellular surface of Cya (site Ex1) leads to profound changes within the intramembrane part of the protein, as well as at a distant catalytic domain.

## Discussion

The cryo-EM structure of *M. tuberculosis* Cya provides a unique insight into the assembly and regulation of a model membrane AC. To this day, the functional role of the TM region in the polytopic

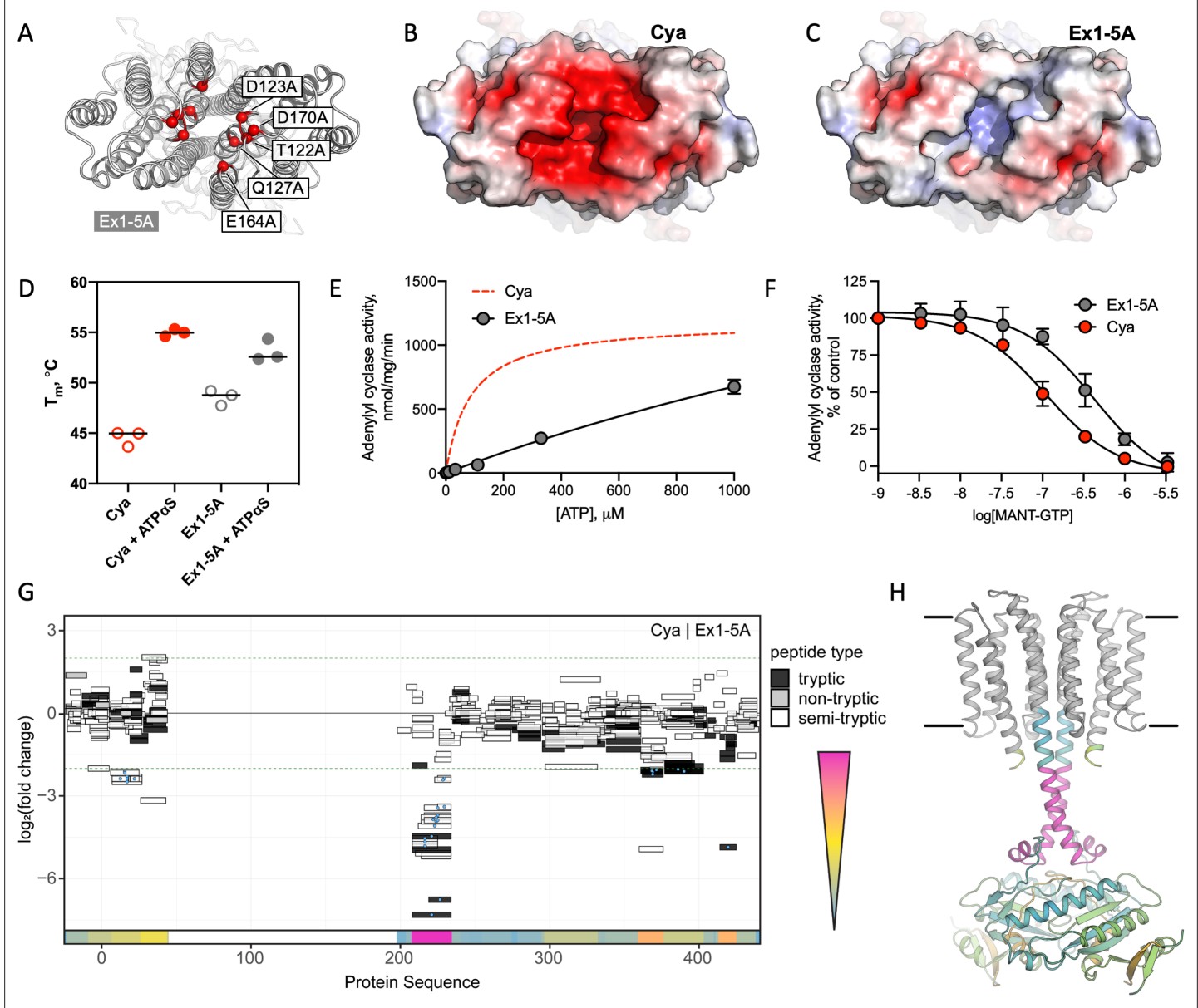

**Figure 5.** Extracellular site Ex1 is linked to the adenylyl cyclase activity of Cya. (**A**) An illustration of the Ex1 site residues mutated to generate the Ex1-5A mutant, substituting the five indicated residues with Ala. (**B**) Calculated electrostatic potential of the wild-type Cya. (**C**) Same as B, for the Ex1-5A mutant. (**D**) The mutant shows thermostability profile consistent with that of the wild-type protein, based on the observed $T_m$ values in the presence and in the absence of a nucleotide analogue. For experiments in D–F, $n = 3$; data are shown as mean ± standard error of the mean (SEM). (**E**) The enzymatic properties of the mutant are substantially affected by the mutation (the dashed red curve corresponds to the fit shown for Cya in **Figure 1D** for comparison). (**F**) The affinity of the Ex1-5A mutant for MANT-GTP is reduced (110 and 420 µM, respectively). (**G**) Limited proteolysis-coupled mass spectrometry (LiP-MS) analysis of Cya and Ex1-5A mutant. The graph indicates sequence coverage and the identified tryptic, semi-tryptic, or non-tryptic peptides. Significantly changing peptides (|log2(FC)|>2; $q$ value <0.001) are marked with a blue dot. A bar within the plot is coloured according to the change in protease accessibility at each peptide (blue = no change, pink = high fold change; absolute log2 transformed fold changes range from 0 to 7.3). (**H**) A model of Cya coloured according to the bar in **E**.

The online version of this article includes the following source data and figure supplement(s) for figure 5:

**Source data 1.** Biophysical and biochemical properties of the Ex1-5A mutant (**Figure 5D–F**).

**Figure supplement 1.** Purification and stability of Cya mutant Ex1-5A.

**Figure supplement 2.** Molecular dynamics (MD) simulations of Cya mutant Ex1-5A.

membrane AC, such as Cya or the mammalian AC1–9, remains elusive. Why does a cell need an AC with such an elaborate membrane anchor? A lipid anchor or a single TM helix would be sufficient to target the catalytic domain to the membrane compartment where cAMP production is required. Our structure provides two possible reasons for the ACs to have such a TM domain: (1) to facilitate the assembly of the HD domain and (2) to act as a receptor module, binding ligands at several newly identified putative ligand-binding sites, including the pockets within the Cya monomers and at the Cya–Cya interface (*Figure 6*). The two functions are not mutually exclusive, as the ligand interactions with the membrane region of the AC may influence the HD assembly and thus regulate the cyclase function. Previous work utilizing chimeric constructs composed of fragments of the quorum sensing receptor CqsS from *Vibrio harveyi* and Cya (*Beltz et al., 2016*) or mammalian membrane ACs (*Seth et al., 2020*) revealed that the membrane anchors of the ACs may act as orphan receptors for yet unknown ligands. Together with the proposed functional coupling between the TM domain and the catalytic site of Cya, the structure described here is consistent with these findings, offering molecular insights into the potential receptor role of the membrane anchor of a model membrane AC.

Our experimental results and simulations point to a possible link between the enzymatic activity of Cya and binding of small cations (such as metals ions) to its Ex1 site. It is possible that transient interactions of cations with protein surfaces play a basic role in surface charge compensation. However, the properties of the Ex1 site are strongly suggestive of a specific ligand-binding site, with a cluster of 10 polar residues (6 negatively charged residues) at the Cya dimer interface pointing towards the Ex1 cavity. The most telling evidence for the role of this site in cation binding is the MD simulation, which revealed two locations on the Cya surface where the metal ions dwell: the established metal-binding site in the catalytic pocket, and the site Ex1. Modification of the charges of the Ex1 by mutagenesis reduced the activity of the protein, further suggesting that this site at the extracellular surface of the protein likely plays a pivotal role in controlling the assembly of the catalytically active dimeric Cya.

It is worth noting that sensing of extracellular metals is well established in eukaryotic and prokaryotic cell signalling. For example, a prominent example in mammalian signalling pathway is the calcium-sensing receptor (CaSR), a GPCR involved in regulation of $Ca^{2+}$ homeostasis (*Gao et al., 2021*). In bacteria, salt sensing has been shown in histidine kinases (*Sphingomonas melonis* KipF) (*Kaczmarczyk et al., 2015*) and in chemotaxis receptors (*E. coli* Tar) (*Bi et al., 2018*). Thus, a potential link between extracellular metal ion sensing and cAMP signalling that may be mediated by the *M. tuberculosis* Cya is within the realm of possibility and presents an interesting avenue for future investigations.

The presence of potential ligand-binding pockets at the extracellular surface of Cya lends strong support to the long-standing idea that the TM domains of the ACs may act as receptor modules for yet unknown ligands (*Beltz et al., 2016*). The possibility of direct regulation of cAMP production via the membrane anchors of the ACs would have long-reaching consequences: a vast repertoire of pharmacological agents on the market today act via GPCRs coupled to membrane ACs (*Sriram and Insel, 2018*). Direct modulation of the cAMP production through the AC membrane domains could revolutionize the approaches to drug development for a wide range of diseases where the GPCRs are currently the primary drug targets. A related notion of importance for molecular pharmacology and medicinal chemistry is the potential interactions between the already existing drugs and the membrane domains of the ACs. Such interactions may lead to unwanted side effects associated with cAMP signalling, such as emesis and changes in heart rate and contractility (*Parnell et al., 2015*). An example of this is the antifungal drug miconazole, which is known to have cardiotoxic effects (*Fainstein and Bodey, 1980*; *Won et al., 2012*). Miconazole has recently been shown to directly activate AC9, likely via its TM domain (*Simpson et al., 2019*). The interaction with AC9 may contribute to the cardiorespiratory side effects of this drug. Similar interactions involving other drug/AC combinations may have to be systematically evaluated, especially in the cases where changes in cAMP levels or in the downstream signalling events are recognized as side effects.

In the absence of known interaction partners for the membrane regions of the ACs it is difficult to predict the effect of ligand binding to any of the pockets we find in the Cya structure. Nevertheless, the structure hints at ways that could be exploited by various agents to affect the AC activity via the membrane domain. The closest example of cyclase modulation through the receptor-mediated effect on the HD helices is the case of soluble GC, described at atomic resolution, has been detailed above (*Figure 3*; *Kang et al., 2019*). The presence of disease-linked mutations in the HD regions of AC5 (*Chen et al., 2012*; *Carecchio et al., 2017*) and retGC1 (*Tucker et al., 2004*; *Zhao et al., 2013*)

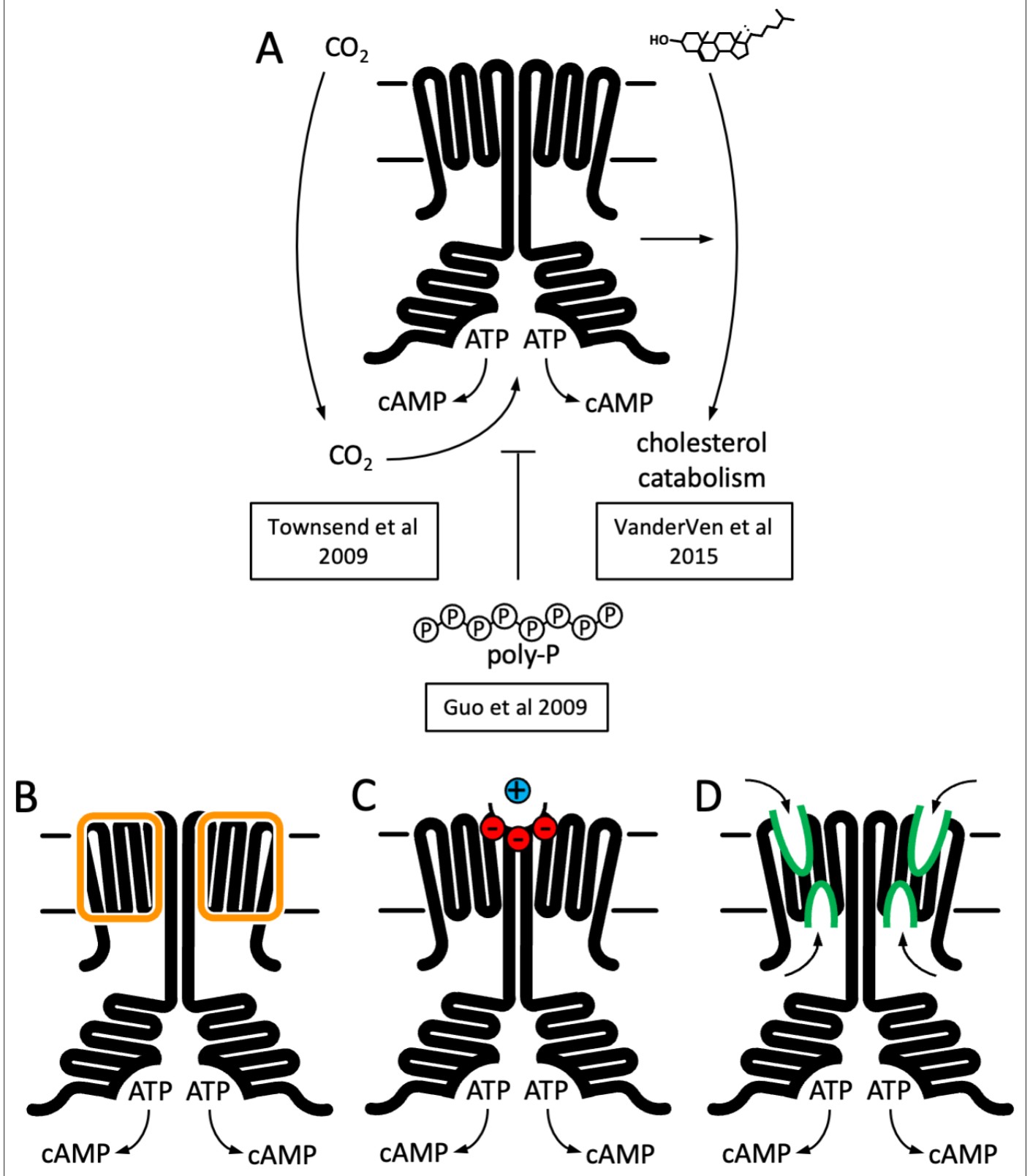

**Figure 6.** Function and structure of Rv1625c/Cya. (**A**) Known regulators and cellular functions of Cya. Insights into the function of the membrane domain of Cya, with new functions of the transmembrane (TM) domain suggested by the cryo-EM structure: a stabilizer of the cytosolic domain assembly (**B**), a receptor for positively charged agents via the Ex1 site at the Cya dimer interface (**C**), and a receptor of yet unknown ligands via sites Ex2/Cy1/Cy2 (**D**).

underscore the importance of this domain for cyclase structure and function. It is clear that the HD region plays a vital part in AC and GC assembly and stability. This is evident from our experiments with the *M. intracellulare Cya* (*Vercellino et al., 2017*), as well as the results of others using Cya and mammalian ACs as model enzymes (*Guo et al., 2001*; *Ziegler et al., 2017*; *Seth et al., 2020*). It remains to be determined whether any agents can elicit conformational changes in the membrane domain of Cya (or in any of the mammalian membrane ACs), leading to substantial changes in the HD similar in scale to the changes observed in the sGC during its activation. Our MD simulations and the experiments with the Ex1-5A mutant of Cya are suggestive of a receptor–transducer–catalyst relay, where the extracellular portion of the TM region acts as a 'receptor' for a yet unknown ligand, the HD transduces the activation, and the catalytic domain catalyses ATP to cAMP conversion. This notion is further supported by previous work on Cya that identified the HD region as a transducer of a putative signal (*Beltz et al., 2016*; *Ziegler et al., 2017*). The structure of Cya can serve as a starting point for exploration of the TM domain-mediated regulation of membrane ACs.

## Materials and methods

### Protein expression and purification

#### Expression and purification of the full-length Cya

Cya cloned into a vector with an N-terminal strep tag and a 3C cleavage tag was expressed in *E. coli* BL21-CodonPlus (DE3)-RIPL cells (Cat. No. 230280, Agilent) grown in TB medium. Protein expression was induced when at $OD_{600}$ of 3.0 using 0.3 mM isopropyl β-d-1-thiogalactopyranoside (IPTG). After 3 hr of induction the cells were harvested. The membranes were prepared using cells lysis by three passes in Emusiflex high pressure homogenizer in a buffer containing 50 mM Tris pH 7.5, 200 mM NaCl, 5 µg/ml DNase, and 1 mM phenylmethylsulfonyl fluoride (PMSF). Lysed cells were centrifuged at 12,000 rpm using a Ti45 rotor for 30 min. The resulting supernatant was spun down by ultracentrifugation using Ti45 rotor at 40,000 rpm for 1 hr, resuspended in a buffer containing 50 mM Tris pH 7.5 and 200 mM NaCl and ultracentrifuged again. The resulting membrane pellet was resuspended in the same buffer, flash frozen and stored at −80°C until purification.

For purification, the membranes were thawed and resuspended in a buffer containing 50 mM Tris pH 7.5, 200 mM NaCl, 10% glycerol, and 1% sol-grade dodecylmaltoside (DDM, Anatrace), mixed at 4°C for 1 hr and ultracentrifuged. The supernatant was incubated with Strep-tactin superflow resin for 1 hr at 4°C. The resin was washed with a volume 25 times that of the resin bed of a buffer containing 0.1% digitonin and the eluted with 5 mM desthiobiotin. The eluted protein was concentrated and injected onto Superose 6 Increase column pre-equilibrated with a buffer containing 50 mM Tris pH 7.5, 200 mM NaCl 0.1% digitonin and 10% glycerol. For cryo-EM samples glycerol was omitted during SEC step.

#### Cya-SOL expression and purification

Cya-SOL construct was generated by cloning the sequence encoding the Cya residues 203–443 into a vector with an N-terminal 10xHis tag followed by a 3C cleavage site. The construct was expressed in *E. coli* BL21-CodonPlus (DE3)-RIPL cells grown in TB medium. Expression was carried out under conditions similar to those used for expression of the full-length protein, with a 5-hr induction at 20°C. The cells were collected by centrifugation, lysed and the cleared lysate was incubated with Ni-NTA resin for 1 hr. The resin was washed with a volume 15 times the resin bed volumne of wash buffer containing 50 mM Tris pH 7.5, 200 mM NaCl, 10% glycerol, and 20 mM imidazole, followed by an additional wash step with a volume of 25 times the volume of the resin bed volume with a buffer containing 50 mM imidazole. The protein was eluted with a buffer containing 250 mM imidazole, concentrated and desalted using a GE PD-10 Sephadex G-25 desalting column. The protein was mixed with 3C protease (1/50, wt/wt) and incubated at 4°C overnight. The protein was passed through pre-equilibrated Ni-NTA resin to remove the 3C protease and purified by SEC using Superdex 200 Increase column.

### Nanobody library generation and selections

To generate desired immune response in heavy chain-only IgG subclass, an alpaca was immunized four times in 2-week intervals, each time with 200 µg purified Rv1625c in phosphate-buffered saline

containing 0.02% (wt/vol)n-Dodecyl-β-D-Maltopyranoside (β-DDM). The antigen was mixed in a 1:1 (vol/vol) ratio with GERBU Fama adjuvant (GERBU Biotechnik GmbH, Heidelberg, Germany) and injected subcutaneously in 100 µl aliquots into the shoulder and neck region. Immunizations of alpacas were approved by the Cantonal Veterinary Office in Zurich, Switzerland (animal experiment licence nr. 172/2014). One week after the last injection, 60 ml of blood was collected from jugular vein for isolation of lymphocytes (Ficoll-Paque PLUS, GE Healthcare Life Sciences, and Leucosep tubes, Greiner). Approx. 50 mio. cells were used to isolate mRNA (RNeasy Mini Kit, Qiagen) that was reverse transcribed into cDNA (AffinityScript, Agilent, USA) using the gene-specific primer. The VhH (nanobody) repertoire was amplified by PCR and phage library was generated by fragment exchange cloning (*Geertsma and Dutzler, 2011*) into a PmlI-linearized pDX phagemid vector. The resulting VhH-phage library (size 4.5 e6) was screened by biopanning against the immobilized target. For that purpose VI23.60 containing Strep-tag was immobilized on the Strep-Tactin coated microplate (IBA Lifesciences GmbH, Germany) and three rounds of selection were performed. One hundred and ninety-five single clones from the enriched nanobody library were induced to express polyhistidine-tagged soluble nanobodies in the bacterial periplasm and analysed by ELISA for binding to the target. Ninety-six ELISA-positive clones were Sanger sequenced and grouped in 17 families according to their CDR3 length and sequence (*Geertsma and Dutzler, 2011*).

## Nanobody expression and purification

Nanobody NB4 was expressed in BL21-CodonPlus (DE3)-RIPL cells in TB medium supplemented with 2 mM magnesium chloride and 0.1% glucose by induction at an $OD_{600}$ of 0.7 using 1 mM IPTG at 26°C for 16 hr. The periplasmic fraction was isolated by resuspending the cell pellet in 2.5× (wt/vol) cold TES buffer (200 mM Tris pH 8.0, 0.5 mM ethylenediaminetetraacetic acid (EDTA) and 0.5 mM sucrose and 1 mM PMSF) for 45 min, followed by an overnight incubation with twice the amount of a fourfold diluted TES buffer. The suspension was spun down and the supernatant was used for protein purification with Ni-NTA resin, following the same procedure as that used for Cya-SOL. The eluted nanobody was concentrated and further purified using SEC with a Superdex 200 Increase column.

## AC activity assay

AC activity assays were performed as described previously (*Vercellino et al., 2017*). In brief, the assay was carried out in a reaction volume of 200 µl with 50 mM Tris pH 8.0, 200 mM NaCl, 5 mM $MgCl_2$, 5 mM $MnCl_2$, and 0.1% digitonin. For determination of $K_m$, ATP concentration was varied from 0 to 1000 mM in the presence of 10 nM [³H]ATP (PerkinElmer). The reaction was initiated by adding ATP (pre-incubated at 30°C for 10 min) to the reaction solution containing 0.005 mg/ml Cya, 0.0075 mg/ml NB4, followed by an incubation for 10 min at 30°C. For Cya-SOL, the reactions were performed in the absence of detergent, using 0.08 mg/ml Cya-SOL, in the presence or in the absence of 0.12 mg/ml NB4. The reaction was stopped by incubating the reaction mixture at 95°C for 4 min and by addition of 20 µl of 2.2 M HCl. The stopped reactions were applied to 1.3 g of aluminum oxide in disposable columns. The cAMP was eluted with 4 ml of 100 mM ammonium acetate into scintillation vials and mixed with 12 ml scintillation liquid (Ultima Gold). The amount of radioactive cAMP was measured using a liquid scintillation counter.

## Isothermal titration calorimetry

The isothermal titration calorimetry (ITC) experiments were performed using a Microcal ITC200 instrument with cell temperature maintained at 25°C and with stirring set to 750 rpm. In total, 15 injections were performed per experiment with each injection set at 2 µl and a pre-injection volume of 0.8 µl. Cya-SOL was kept in the cell at a concentration of 30 µM and NB4 was kept in the syringe at 300 µM. All ITC measurements were performed in triplicates. The results were analysed using sedphat and NITPIC. The figures describing the ITC results were generated using GUSSI (*Brautigam et al., 2016*).

## Protein thermal unfolding

Protein thermal stability was measured using nanoDSF on a Prometheus panta instrument (NanoTemper) (*Magnusson et al., 2019*). The protein was measured at 0.5 mg/ml concentration in a buffer containing 50 mM Tris pH 7.5, 200 mM NaCl, and 0.1% digitonin, using NT.48 capillaries. For samples with ligand, 1 mM ATPαS and 5 mM $MgCl_2$ was added and allowed to incubate at RT for 10 min.

The samples were spun at 13,000 × $g$ on a tabletop centrifuge for 1 min before measuring. Thermal unfolding experiments were carried out at a temperature increment of 1°C/min in triplicates. $T_m$ was calculated as the first derivative of intrinsic protein emission ratio at 350 and 330 nM using PR.Panta analysis software.

## Cryo-EM sample preparation

For cryo-EM sample preparation, freshly purified full-length Cya in 0.1% digitonin was concentrated and mixed with NB4 at a molar ratio of 1:1.5. Additionally, 5 mM $MnCl_2$ and 0.5 mM MANT-GTP were added and the mixtures were incubated on ice for 30 min. The final concentration of Cya was 5–6 mg/ml. An aliquot of 3.5 µl of sample was placed on the glow-discharged cryo-EM grid (Quanti-foil R1.2/1.3 or Quantifoil R2/1), blotted and plunge frozen in liquid ethane using a Mark VI Vitrobot instrument maintained at 100% humidity with blot force 20 and blot time of 3 s. The grids were cryo-transferred for storage in liquid nitrogen.

## Cryo-EM data acquisition and image analysis

The cryo-EM data were obtained at the SCOPEM facility at ETHZ using a 300 kV Titan Krios electron microscope (FEI) equipped with a K3 direct electron detector with a pixel size of 0.33 Å/pix (in super-resolution mode), at a defocus range of −0.5 to −3.0 µm. All movies were dose fractionated into 40 frames. The movies for dataset 1 were recorded with a total dose of 54 e⁻/Å², dataset 2 – with a dose of 47 e⁻/Å², and for dataset 3 a dose of 44 e⁻/Å².

All data processing was performed in relion 3.0 (*Zivanov et al., 2018*). All micrographs were motion corrected using motioncorr 1.2.0 (*Zheng et al., 2017*) and binned twofold. All micrographs were CTF corrected using Gctf (*Zhang, 2016*). Particles were autopicked using templates from manual picking. In total 1,692,104 particles were picked for dataset 1, 1,898,968 particles for dataset 2, and 990,286 particles for dataset 3. After several rounds of 2D classification, datasets 1, 2, and 3 were left with 253,789, 1,173,076, and 741,081 particles, respectively. 3D classification with four classes was used to further process each dataset, with C1 and C2 symmetry imposed. The particles from the best classes in each dataset were chosen and further refined. The extracellular density for NB4 was masked out for all subsequent refinements to generate refined 3D maps at a resolution of 4.37 Å (dataset 1), 4.25 Å (dataset 2), and 4.61 Å (dataset 3) in C2 symmetry. The particles were merged into a single selection and subjected to refinement, ctf refinement and particle polishing, yielding a final refined map of 3.57 Å resolution (C2 symmetry). The same particle selection produced a reconstruction at 3.83 Å resolution without symmetry imposed (C1). Model building was performed in coot (*Emsley and Cowtan, 2004*). The model was refined using phenix.real_space_refine in Phenix (*Adams et al., 2010*). For model validation, the model atoms were randomly displaced (0.5 Å), and the resulting model refined using one of the refined half maps (half-map1). Map versus model FSC was calculated using the model against the corresponding half-map1 used for refinement, and for the same model versus the half-map2 (not used in refinement) (*Amunts et al., 2014*). Model geometry was assessed using MolProbity (*Chen et al., 2010*).

## Protein crystallization, X-ray data collection, processing, and structure determination

Crystallization of Cya-SOL–NB4 complex was performed using standard vapour diffusion techniques at 20°C. Concentrated protein complex was prepared by mixing purified Rv1625c and NB4 in 1:1.2 ratio in a buffer of the following composition: 20 mM Tris–HCl pH 7.5, 150 mM NaCl, 5 mM $MnCl_2$, and 1 mM MANT-GTP. This protein solution was used to set up 96-well sitting drop crystallization trials using TPP Mosquito LCP robot. Formulatrix Rockimager was used to visualize crystal formation. Multiple crystal hits were obtained, and selected conditions were used as starting points for further crystal optimizations. The optimal crystals were obtained after mixing 1.5 µl of protein (20 mg/ml) with 1.5 µl of reservoir solution (0.1 M Na-acetate pH 5.5, 0.02 M $CaCl_2$, 30% 2-methyl-2,4-pentanediol (MPD)). The crystals were gently transferred to cryoprotectant consisting of 0.05 M Na-acetate pH 5.5, 0.01 M $CaCl_2$, 35% MPD, and crystals were subsequently mounted onto crystal loop (Hampton Research) and flash frozen in liquid nitrogen.

X-ray data collection was performed at the PXI and PXIII beamlines at the Swiss Light Source synchrotron in Villigen, Switzerland. The best dataset was collected to 1.97 Å resolution from a single

crystal at cryogenic temperature (100 K) using Eiger detector with oscillation range of 0.1°. The data were processed using MOSFLM and XDS (*Kabsch, 2010*; *Leslie and Powell, 2007*). The resolution cutoff was chosen taking into account the values of CC1/2 and mean I/sigma(I) (*Karplus and Diederichs, 2012*). Phasing/refinement was performed using Phenix (*Adams et al., 2010*). Phases were resolved by molecular replacement using templates (cya: 5O5K, nanobody: 6FPV). Coot was used for model building and geometrical optimization (*Emsley and Cowtan, 2004*). Crystallographic data collection and refinement statistics are shown in *Supplementary file 2*.

## MD simulations

The MD simulations were performed using CHARMM36m force field in GROMACS 2019.3 (*Kutzner et al., 2019*). The missing N- and C-terminus of Cya were modelled using I-TASSER server (*Yang et al., 2015*), and the protein was inserted into lipid membrane, solvated and ionized using the Membrane Builder tools in CHARMM-GUI (*Jo et al., 2008*). Lipid membrane composed of 60 POPC, 40 POPG, 40 POPE, 20 POPI, 20 DLGL, and 20 cholesterol molecules. The system was solvated with TIP3 water molecules extended to 25 Å from the edge of the protein and with per lipid hydration number of 50. Subsequently the system was neutralized with $Cl^-$ ions, and then was brought to a final concentration of 0.15 M KCl or 0.15 M $MgCl_2$. All of the generated systems were subjected to energy minimization, and six-step NVT and NPT equilibration using the default scheme provided in the CHARM-GUI (*Halls and Cooper, 2011*), followed by 200 ns of a production run. Final trajectories were analysed using tools in available the GROMACS package. The Volmap tool in VMD (*Humphrey et al., 1996*) was used for generating occupancy/density maps. The figures related to the MD simulations were prepared using VMD or Pymol.

## Limited proteolysis-coupled mass spectrometry

Wild-type and mutant Cya protein preparations were diluted in LiP buffer (1 mM $MgCl_2$, 150 mM KCl, 100 mM N-2-hydroxyethylpiperazine-N'-2-ethanesulfonic acid (HEPES)–KOH pH 7.4) with 0.1% digitonin. Mutant and wild-type samples were split into eight samples each at a protein amount of 2 µg of protein per 50 µl of buffer. Four out of eight WT and mutant samples were treated with PK from *Tritirachium album* (Sigma-Aldrich) (limited proteolysis, LiP), whereas the other four were treated with water instead (TC). The samples were incubated in a Thermocycler for 5 min at 25°C. PK was inactivated by heating the samples to 99°C for 5 min, then incubating them at 4°C for 5 min, followed by the addition of the same volume of 10% sodium deoxycholate. LiP and TC underwent the same procedures.

### Tryptic digest

Following the addition of sodium deoxycholate, disulfide bonds were reduced by adding tris(2-carboxyethyl)phosphine to a final concentration of 5 mM and incubating the samples at 37°C for 40 min with slight agitation. Free cysteine residues were alkylated with iodoacetamide at a final concentration of 40 mM and 30 min of incubation at room temperature in the dark with slight agitation. The samples were diluted with 100 mM ammonium bicarbonate to a final sodium deoxycholate concentration of 1%. Lysyl endopeptidase LysC was added at an enzyme to substrate ratio of 1:50 and samples were incubated for 1 hr at 37°C with slight agitation. Next, trypsin was added at an enzyme to substrate ration of 1:50 and incubated at 37°C overnight with slight agitation. The digestion was stopped by adding 50% formic acid to the samples to achieve a final concentration of 2% formic acid. Precipitated sodium deoxycholate was removed by filtering through a Corning 2 µM polyvinylidene difluoride (PVDF) plate and samples were further desalted on a 96-well MacroSpin plate (The Nest Group). Peptides were eluted with 80% acetonitrile, 1% formic acid and dried in a vacuum centrifuge. After drying, samples were reconstituted in 20 µl 0.1% formic acid and iRT peptides (Biognosys) were added.

### LC–MS/MS data acquisition

Samples were measured on an Orbitrap Exploris 480 mass spectrometer (Thermo Fisher), equipped with a nanoelectrospray source and an Easy-nLC 1200 nano-flow LC system (Thermo Fisher). 1 µl of digest was injected and separated on a 40 cm × 0.75 i.d. column packed in-house with 1.9 µm C18 beads (Dr. Maisch Reprosil-Pur 120) using a linear gradient from 3% to 35% B (eluent A: 0.1% formic

acid, eluent B: 95% acetonitrile, 1% formic acid). Gradient duration was 30 min, whereas the whole method was 60 min long. Samples were measured at a constant flowrate of 300 nl/min while the column was heated to 50°C. All samples were acquired in DIA (41 windows, 1 $m/z$ overlap) and analysed in Spectronaut v15 (Biognosys). Further data analysis was carried out in R using mainly the R package protti (**Quast, 2021**). Briefly, the abundances of Rv1625c mutant and wild type were compared in the tryptic controls and the protein abundances in the LiP samples were corrected accordingly. Statistical testing on peptide level to detect peptide abundance differences was conducted employing the proDA (**Ahlmann-Eltze, 2020**) algorithm, implemented in protti. The Rv1625c PDB file was edited and the $B$ factors were replaced with the maximum absolute value of the calculated $\log_2$(fold change) at each position. In pyMOL, the protein was then coloured according to the replaced $B$ factors to highlight regions changing regions.

## LiP-MS data interpretation

Whether a peptide decreases or increases in abundance is dependent on the accessibility of the native protein to PK. In a standard LiP-MS experiment, three peptide types can be detected, dependent on the proteolytic cleavage:

- Semi-tryptic peptides are generated by a cleavage of PK on either the N-terminal or the C-terminal side of the peptide and a cleavage by trypsin on the respective other side.
- Tryptic peptides are not cleaved by PK at all.
- Non-tryptic peptides were cleaved by PK on both sides.

Depending on the peptide type an increase or decrease in abundance can be interpreted in different ways. A tryptic peptide that decreases in abundance was additionally cleaved by PK, hence it disappears. This likely means that the protein region became more accessible to PK. On the other hand, a tryptic peptide that increases in abundance can be interpreted as the region becoming less accessible to PK. A semi-tryptic peptide that increases in abundance can be explained as the protein region cleaved by PK becoming more accessible. A semi-tryptic peptide that decreases in abundance can be explained in two different ways: either the protein region became more protected, hence inaccessible to PK, or the protein region became more accessible and the peptide was not detected because of additional PK cleavage sites that were introduced with the conformational change.

## Acknowledgements

We thank Emiliya Poghosyan and Elisabeth Müller-Gubler (EM Facility, PSI), and Miroslav Peterek (ScopeM, ETH Zurich) for their support in cryo-EM data collection. We also thank Spencer Bliven and Marc Caubet Serrabou (PSI) for support in high performance computing. Funding: Swiss National Science Foundation (150665; VMK), Swiss National Science Foundation (176992; VMK), Swiss National Science Foundation (184951; VMK), and Vontobel Stiftung (VMK).

## Additional information

### Funding

| Funder | Grant reference number | Author |
| --- | --- | --- |
| Schweizerischer Nationalfonds zur Förderung der Wissenschaftlichen Forschung | 150665 | Volodymyr M Korkhov |
| Schweizerischer Nationalfonds zur Förderung der Wissenschaftlichen Forschung | 176992 | Volodymyr M Korkhov |

| Funder | Grant reference number | Author |
|---|---|---|
| Schweizerischer Nationalfonds zur Förderung der Wissenschaftlichen Forschung | 184951 | Volodymyr M Korkhov |

The funders had no role in study design, data collection, and interpretation, or the decision to submit the work for publication.

### Author contributions

Ved Mehta, Conceptualization, Data curation, Validation, Investigation, Visualization, Methodology, Writing – original draft, Writing – review and editing; Basavraj Khanppnavar, Conceptualization, Data curation, Formal analysis, Validation, Investigation, Visualization, Methodology, Writing – original draft, Writing – review and editing; Dina Schuster, Conceptualization, Data curation, Formal analysis, Validation, Investigation, Visualization, Methodology, Writing – review and editing; Ilayda Kantarci, Angela Kosturanova, Tarun Iype, Investigation, Methodology; Irene Vercellino, Conceptualization, Formal analysis, Investigation, Methodology; Sasa Stefanic, Data curation, Validation, Investigation, Methodology; Paola Picotti, Resources, Supervision, Writing – review and editing; Volodymyr M Korkhov, Conceptualization, Resources, Data curation, Formal analysis, Supervision, Funding acquisition, Validation, Investigation, Visualization, Methodology, Writing – original draft, Project administration, Writing – review and editing

### Author ORCIDs

Dina Schuster (iD) http://orcid.org/0000-0001-6611-8237
Sasa Stefanic (iD) http://orcid.org/0000-0001-7367-1831
Volodymyr M Korkhov (iD) http://orcid.org/0000-0002-0962-9433

### Decision letter and Author response

Decision letter https://doi.org/10.7554/eLife.77032.sa1
Author response https://doi.org/10.7554/eLife.77032.sa2

## Additional files

### Supplementary files

- Supplementary file 1. Cryo-EM analysis and statistics.
- Supplementary file 2. X-ray data analysis and statistics.
- Transparent reporting form

### Data availability

The atomic coordinates and structure factors have been deposited in the Protein Data Bank (7YZ9, 7YZI, 7YZK); the density maps have been deposited in the Electron Microscopy Data Bank (EMD-14388, EMD-14389). The mass spectrometry data have been deposited to the ProteomeXchange Consortium via the PRIDE partner repository with the dataset identifier PXD033826. All other data are available in the main text or the supplementary materials.

The following dataset was generated:

| Author(s) | Year | Dataset title | Dataset URL | Database and Identifier |
|---|---|---|---|---|
| Schuster D, Korkhov VM | 2022 | Structural impact of TM domain mutations on mycobacterial adenylyl cyclase Rv1625c (LiP-MS) | https://www.ebi.ac.uk/pride/archive/projects/PXD033826 | PRIDE, PXD033826 |

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
