## [Editor Report]

This manuscript reports the first full-length structure of membrane-bound adenylyl cyclase from the pathogen *Mycobacterium tuberculosis*. The structure provides insights into its potential mechanism of action and reveals similarities to its mammalian counterpart. Thus, this paper is of potential interest to a broad audience including the fields of infectious diseases, signaling, and evolutionary biologists.

---

## [Decision Letter]

**Decision letter after peer review:**

Thank you for submitting your article "Structure of *Mycobacterium tuberculosis* Cya, an evolutionary ancestor of the mammalian membrane adenylyl cyclases" for consideration by *eLife*. Your article has been reviewed by 2 peer reviewers, one of whom is a member of our Board of Reviewing Editors, and the evaluation has been overseen by Nancy Carrasco as the Senior Editor. The reviewers have opted to remain anonymous.

Essential revisions:

1) The manuscript would benefit from a more detailed comparison with the AC9 structure. In particular, are the four potential pockets in Cya also found in AC9?

2) It would be helpful to comment and show a figure of the conservation of the pockets. Are they conserved both among similar mycobacterial enzymes and the mammalian family? Does this tell the authors anything about their role (i.e. are they different, suggesting different ligand binding sites, or the same suggesting a similar compound could be bound)?

3) The authors suggest site Ex1 binds a metal and that this may play a sensing role. if the authors make this claim I think they should provide evidence that the occupancy of the site can change under the expected physiological concentration changes. Can MD provide some hints to the metal affinity? Can changing buffers or adding chelating agents remove the ion? Is there any evidence for extracellular ion sensors?

4) The authors show in Figure 1, that SOL Cya (203-433) is not active. It binds NB4 with high affinity but no mention of its effect on activity. A data set of SOL-NB4 should be added to panel 1D for completeness. Moreover, the authors should contrast their results with a previous report (Guo, 2001) showing a similar construct [MRGSH6GS-Cya (204-443)] fully active (sp. act. ~ 1-2 μmol cAMP/mg/min) as a dimer.

*Reviewer #1 (Recommendations for the authors):*

The authors show in Figure 1, that SOL Cya (203-433) is not active. It binds NB4 with high affinity but no mention of its effect on activity. A data set of SOL-NB4 should be added to panel 1D for completeness. Moreover, the authors should contrast their results with a previous report (Guo, 2001) showing a similar construct [MRGSH6GS-Cya (204-443)] fully active (sp. act. ~ 1-2 μmol cAMP/mg/min) as a dimer.

The scheme in Figure 1C indicates the soluble cyclase includes residues 203-443; however, the methods section indicates "cloning the sequence encoding the Cya residues 203-433". Which one is correct?

The authors provide an explanation for previous biochemical data indicating a role for positively charged R43-R44 (mutants inactive) in the N-ter stabilizing the negatively charged surface of HD. Does addition in trans of an N-terminus-derived peptide (containing residues V41ARRQR46) affect SOL activity?

The topological organization of the TM bundles of Cya and AC9 is remarkably conserved. However, the structure revealed a difference in orientation of the corresponding TM regions in Cya vs. AC9 relative to the catalytic domains (90 degrees rotation) that the authors claim is due to the difference between the HD lengths. Does the data rule out a potential "non-specific" effect of the third binding site for NB4?

The authors argue that the conserved TM1-5 in Cya and corresponding TM1-5/TM7-11 in AC9 act as a guide allowing the TM6 to communicate to the HD and catalytic core. The organization indicates that TM4-5-6 might be sufficient to fulfill this role. Is a full-length Cya with deletion of TM1-2 [N-TM (∆TM1-2)-HD-cat] active?

The structure revealed a pocket (Ex1) formed at the extracellular interface of the two TM bundles lined by the acidic residues D123, E164, and D170 of each monomer. An unassigned density is observed, and the authors suggest the potential involvement of a positively charged molecule as a putative ligand.

Interestingly, a similar pocket is present in AC9. Is there any equivalent density present in the mammalian AC9 structure previously reported by the same group?

Moreover, this is the binding for the off-target third NB4 site. Can the authors rule out this "ghost density" as part of the NB4 CDR loops? If the spurious binding of NB4 also utilizes these negative charges, shouldn't NB4 and putative ligand (ghost density) be mutually exclusive?

Since the paper title provocative suggest Cya as an evolutionary ancestor of the mammalian membrane adenylyl cyclases, it will benefit the readers to include a paragraph (albeit speculative) of what the authors (and the field) think of how the bacterial homodimeric cyclase with two catalytic domains evolved in mammals into a heterodimeric (C1 and C2) form with a single catalytic domain and a second allosteric site (occupied by the non-natural AC activator forskolin).

*Reviewer #2 (Recommendations for the authors):*

1) The manuscript would benefit from a more detailed comparison with the AC9 structure. In particular, are the four potential pockets in Cya also found in AC9?

2) It would be helpful to comment and show a figure of the conservation of the pockets. Are they conserved both among similar mycobacterial enzymes and the mammalian family? Does this tell the authors anything about their role (i.e. are they different, suggesting different ligand binding sites, or the same suggesting a similar compound could be bound)?

3) The authors suggest site Ex1 binds a metal and that this may play a sensing role. if the authors make this claim I think they should provide evidence that the occupancy of the site can change under the expected physiological concentration changes. Can MD provide some hints to the metal affinity? Can changing buffers or adding chelating agents remove the ion? Is there any evidence for extracellular ion sensors?

---

## [Author Response]

Essential revisions:1) The manuscript would benefit from a more detailed comparison with the AC9 structure. In particular, are the four potential pockets in Cya also found in AC9?

The Figure S9 shows the comparison of the cavities in AC9 and in Cya side by side. The four cavities are not found in AC9, at least not at the current resolution. We have added additional text to reflect this.

2) It would be helpful to comment and show a figure of the conservation of the pockets. Are they conserved both among similar mycobacterial enzymes and the mammalian family? Does this tell the authors anything about their role (i.e. are they different, suggesting different ligand binding sites, or the same suggesting a similar compound could be bound)?

In the revised manuscript we have included a new figure S12, which illustrates the conservation of sequence in the Ex1 and Ex2 cavities (analysis performed using the ConSurf server with a multiple sequence alignment of 180 mycobacterial homologues of Cya).

3) The authors suggest site Ex1 binds a metal and that this may play a sensing role. if the authors make this claim I think they should provide evidence that the occupancy of the site can change under the expected physiological concentration changes. Can MD provide some hints to the metal affinity? Can changing buffers or adding chelating agents remove the ion?

This is an extremely challenging problem to address experimentally at the moment, because to determine the occupancy changes at the extracellular site we would need to isolate the intracellular site. We are very keen to do this using, for example, liposome-reconstituted purified Cya – subjecting the samples to single particle cryo-EM or cryo-ET and subtomogram averaging. These approaches will be the focus of the future studies in our laboratory. Due to the complexity and technical challenges these experiments fall outside the scope of the current study.

However, to characterize the properties of the Ex1 site, we have performed MD simulations with the Ex15A mutant. The results are shown in the new Figure S13. Mutation of the site abolishes cation binding, without perturbing the canonical metal binding site in the ATP binding pocket of the protein. The description and our interpretation of the observations is on line 243:

“To further support our insights derived from the experiments with the Ex1-5A mutant we performed the MD simulations with Ex1-5A protein (Figure S13), using the identical protocol as those used for the wild-type Cya (Figure S10). The simulation revealed a dramatic difference between the B-factors of the Ex1-5A and the wild-type protein, indicative of increased conformational flexibility of the mutant (Figure S13A). As expected, metal ion binding was preserved at the cytosolic domain of the Ex1-5A, but was completely disrupted at the extracellular Ex1 site (Figure S13B-C). Analysis of the distances between select residues in Cya and in Ex15A showed that although the core remains relatively stable for both protein, the TM1-5 region may be more conformationally flexible (judged by the increased average distances between the residues A77-A77 and L92-L92 within the dimer; Figure S13D-E). The same is true for portions of the catalytic domain, evident from the increased average distance in the residue Q289. Together with the functional data, the results of the MD simulations strongly suggest that disruption of the extracellular surface of Cya (site Ex1) leads to profound changes within the intramembrane part of the protein, as well as at a distant catalytic domain.”

Is there any evidence for extracellular ion sensors?

The extracellular ion sensors indeed exist. In mammals a prominent example is the calcium sensing receptor (CaSR), a G-protein coupled receptor. In bacteria, salt sensing has been shown in histidine kinases (Sphingomonas melonis KipF, Kaczmarczyk et al., 2015; doi: 10.1128/JB.00019-15) or in chemotaxis receptors (*Escherichia coli* Tar, Shuangyu et al., 2018; doi: 10.1038/s41467-018-05335-w). We have added this as a point in the discussion (line 289):

“It is worth noting that sensing of extracellular metals is well established in eukaryotic and prokaryotic cell signaling. For example, a prominent example in mammalian signaling pathway is the calcium-sensing receptor (CaSR), a GPCR involved in regulation of ca^2+^ homeostasis (29). In bacteria, salt sensing has been shown in histidine kinases (Sphingomonas melonis KipF) (30) and in chemotaxis receptors (*Escherichia coli* Tar) (31). Thus, a potential link between extracellular metal ion sensing and cAMP signaling that may be mediated by the *M. tuberculosis* Cya is within the realm of possibility and presents an interesting avenue for future investigations.”

4) The authors show in Figure 1, that SOL Cya (203-433) is not active. It binds NB4 with high affinity but no mention of its effect on activity. A data set of SOL-NB4 should be added to panel 1D for completeness.

We have included the SOL-NB4 dataset for completeness in the Figure 1, as requested. There is no effect on the activity of the protein, as expected based on the assays performed using the full-length Cya.

Moreover, the authors should contrast their results with a previous report (Guo, 2001) showing a similar construct [MRGSH6GS-Cya (204-443)] fully active (sp. act. ~ 1-2 μmol cAMP/mg/min) as a dimer.

Our previous work on the M. intracellulare homologue of Cya has shown that in the absence of the TM domain the soluble domain does not effectively assemble, and in the absence of the helical domain it assembles even less readily (Vercellino et al., 2017). The reason for this is that the ability of the catalytic domain to dimerize diminishes in the absence of the helical domain – but is fully restored in the presence of a non-cyclizable nucleotide analogue MANT-GTP. We could show this using analytical ultracentrifugation and activity assays in our 2017 study. Consistent with those observations, removing the TM domain leads to reduced activity of the SOL domain in the *M. tuberculosis* Cya.

The difference in the catalytical activity observed previously by Guo et al., is likely due to the different conditions used in the assays: the previous study included 22% glycerol, 50 mM Tris pH 7.5, 850 μm MnATP, 2 mM cAMP. In contrast, our assays included no added glycerol and the reactions included 50 mM Tris pH 8.0, 200 mM NaCl, 5 mM MgCl2, 5 mM MnCl_2_ (and 0.1% digitonin for the full length protein). Thus, the most likely explanation for the difference in the observed values is the presence of glycerol and the absence of salt in the Guo et al., experiments – these conditions would strongly favour catalytic domain dimerization. Our conditions we motivated by the need for a direct comparison between SOL and full length Cya. The purpose of the assays in the Guo et al., study was somewhat different (the comparison there was between the SOL domain and the concatenated SOL domain). Furthermore, the activity of the full-length Cya was not reported in the Guo study. In the absence of this control we can not conclude that the SOL domain was fully active in their experimental conditions. The low activity of the SOL domain that we observe is consistent with what we know based on Vercellino et al., 2017, work, and we have a simple explanation for the different values obtained in our hands and in the Guo study. We have added a line explaining this discrepancy very briefly (line 88):

“Previously reported SOL activity values, obtained under different experimental conditions in the presence of low salt and 22% glycerol, showed concentration dependence of the enzymatic activity, consistent with the requirement of the SOL domain to dimerize to form a functionally-active unit (10).”

Reviewer #1 (Recommendations for the authors):The authors show in Figure 1, that SOL Cya (203-433) is not active. It binds NB4 with high affinity but no mention of its effect on activity. A data set of SOL-NB4 should be added to panel 1D for completeness.

We have included the SOL-NB4 dataset for completeness in the Figure 1, as requested. There is no effect on the activity of the protein, as expected based on the assays performed using the full-length Cya.

Moreover, the authors should contrast their results with a previous report (Guo, 2001) showing a similar construct [MRGSH6GS-Cya (204-443)] fully active (sp. act. ~ 1-2 μmol cAMP/mg/min) as a dimer.

Our previous work on the M. intracellulare homologue of Cya has shown that in the absence of the TM domain the soluble domain does not effectively assemble, and in the absence of the helical domain it assembles even less readily (Vercellino et al., 2017). The reason for this is that the ability of the catalytic domain to dimerize diminishes in the absence of the helical domain – but is fully restored in the presence of a non-cyclizable nucleotide analogue MANT-GTP. We could show this using analytical ultracentrifugation and activity assays in our 2017 study. Consistent with those observations, removing the TM domain leads to reduced activity of the SOL domain in the *M. tuberculosis* Cya.

The difference in the catalytical activity observed previously by Guo et al., is likely due to the different conditions used in the assays: the previous study included 22% glycerol, 50 mM Tris pH 7.5, 850 μm MnATP, 2 mM cAMP. In contrast, our assays included no added glycerol and the reactions included 50 mM Tris pH 8.0, 200 mM NaCl, 5 mM MgCl2, 5 mM MnCl_2_ (and 0.1% digitonin for the full length protein). Thus, the most likely explanation for the difference in the observed values is the presence of glycerol and the absence of salt in the Guo et al., experiments – these conditions would strongly favour catalytic domain dimerization. Our conditions we motivated by the need for a direct comparison between SOL and full length Cya. The purpose of the assays in the Guo et al., study was somewhat different (the comparison there was between the SOL domain and the concatenated SOL domain). Furthermore, the activity of the full-length Cya was not reported in the Guo study. In the absence of this control we can not conclude that the SOL domain was fully active in their experimental conditions. The low activity of the SOL domain that we observe is consistent with what we know based on Vercellino et al., 2017, work, and we have a simple explanation for the different values obtained in our hands and in the Guo study. We have added a line explaining this discrepancy very briefly (line 88):

“Previously reported SOL activity values, obtained under different experimental conditions in the presence of low salt and 22% glycerol, showed concentration dependence of the enzymatic activity, consistent with the requirement of the SOL domain to dimerize to form a functionally-active unit (10).”

The scheme in Figure 1C indicates the soluble cyclase includes residues 203-443; however, the methods section indicates "cloning the sequence encoding the Cya residues 203-433". Which one is correct?

We thank the reviewer for spotting the typo – 443 is correct and we have fixed this in the methods.

The authors provide an explanation for previous biochemical data indicating a role for positively charged R43-R44 (mutants inactive) in the N-ter stabilizing the negatively charged surface of HD. Does addition in trans of an N-terminus-derived peptide (containing residues V41ARRQR46) affect SOL activity?

This is an intriguing idea – we will definitely consider such experiments in the future. As the focus of our study was almost entirely on the full-length protein, we have not considered performing such experiments. In a broader context investigating the effects of N-termini on HD assembly will be very interesting – particularly in the case of the mammalian membrane ACs, but also in the case of Cya.

The topological organization of the TM bundles of Cya and AC9 is remarkably conserved. However, the structure revealed a difference in orientation of the corresponding TM regions in Cya vs. AC9 relative to the catalytic domains (90 degrees rotation) that the authors claim is due to the difference between the HD lengths. Does the data rule out a potential "non-specific" effect of the third binding site for NB4?

We do not anticipate an appreciable non-specific effect of the third nanobody on the protein assembly. Our cryo-EM data indicated that the density of the third nanobody is quite poor, consistent with weak binding. A low affinity interaction is likely to be very superficial, not manifesting in any substantial changes in protein assembly. Furthermore, the membrane protein only experiences the nanobody shortly before plunge-freezing into liquid ethane. Thus, at the folding/assembly stage (in the cell) the protein does not encounter the nanobody. This consideration excludes the risk of affecting the fold of the protein. Perhaps the best indication for the lack of any effect of the third nanobody is the lack of any appreciable effect of NB4 on the catalytic activity of the full-length Cya, as well as on the SOL domain.

The authors argue that the conserved TM1-5 in Cya and corresponding TM1-5/TM7-11 in AC9 act as a guide allowing the TM6 to communicate to the HD and catalytic core. The organization indicates that TM4-5-6 might be sufficient to fulfill this role. Is a full-length Cya with deletion of TM1-2 [N-TM (∆TM1-2)-HD-cat] active?

This is an interesting idea, and we will consider exploring it in the future. Defining the minimal unit necessary for correct HD assembly may be very informative. We chose to refrain from performing such experiments within the framework of this first study, primarily because of the scale of the work that would need to be done: we would have to construct a large number of constructs and study each functionally and structurally, ideally not only with the Cya, but also with the mammalian membrane proteins. We estimate this to be a full-scale project for ~1-3 years, outside the scope of our current investigation.

The structure revealed a pocket (Ex1) formed at the extracellular interface of the two TM bundles lined by the acidic residues D123, E164, and D170 of each monomer. An unassigned density is observed, and the authors suggest the potential involvement of a positively charged molecule as a putative ligand.Interestingly, a similar pocket is present in AC9. Is there any equivalent density present in the mammalian AC9 structure previously reported by the same group?

Actually, the quality of the AC9 density map in the extracellular region does not allow us to make any meaningful comparisons with the corresponding region in Cya. We can not claim that there is a similar pocket in AC9, comparable to Ex1 in Cya, because some of the AC9 extracellular loops are not resolved. In this respect, the Cya density map is currently the best available representation of the extracellular surface of a membrane cyclase observed experimentally. The comparisons to the experimentally determined AC9 structures are limited by the quality of the AC9 reconstructions in the extracellular part of the protein. Comparisons to the AlphaFold models of AC9 or other ACs are very complicated, because each of the ACs contains predicted extended extracellular loops, which are very different from those in Cya. As the high resolution cryo-EM density maps of the mammalian ACs are not yet available, in this study we opted to refrain from making direct detailed comparisons to the mammalian ACs.

Moreover, this is the binding for the off-target third NB4 site. Can the authors rule out this "ghost density" as part of the NB4 CDR loops? If the spurious binding of NB4 also utilizes these negative charges, shouldn't NB4 and putative ligand (ghost density) be mutually exclusive?

We can rule out possibility of the ghost density as part of NB4: the density is located deep in the cavity and the CDR loops of the nanobody are located quite far from this site.

Since the paper title provocative suggest Cya as an evolutionary ancestor of the mammalian membrane adenylyl cyclases, it will benefit the readers to include a paragraph (albeit speculative) of what the authors (and the field) think of how the bacterial homodimeric cyclase with two catalytic domains evolved in mammals into a heterodimeric (C1 and C2) form with a single catalytic domain and a second allosteric site (occupied by the non-natural AC activator forskolin).

Although we used the previously published suggestions of an evolutionary relationship between Rv1625c/Cya and the mammalian cyclases as a justification for our manuscript’s title, the focus of our work was not on protein evolution. The possible scenario for mammalian pseudoheterodimeric AC evolution could involve a gene duplication event whereby two half-cyclases (similar to Cya) fused to form a pseudoheterodimer. This would have to be accompanied by a loss of one of the active sites, as we observe only one active site in the mammalian enzymes, with a second site acting as an allosteric activator (forskolin) binding site. The manuscript includes the reference to the seminal study that addressed this directly: ref. 10 (Guo et al., EMBOJ, 2001) and ref. 11 (Guo et al., Mol Microbiol, 2005).

Reviewer #2 (Recommendations for the authors):1) The manuscript would benefit from a more detailed comparison with the AC9 structure. In particular, are the four potential pockets in Cya also found in AC9?

We have revised the Figure S9, and provided a more detailed comparison within the figure panels. The new figure features a more careful comparison of the two structures, with clear labeling.

2) It would be helpful to comment and show a figure of the conservation of the pockets. Are they conserved both among similar mycobacterial enzymes and the mammalian family? Does this tell the authors anything about their role (i.e. are they different, suggesting different ligand binding sites, or the same suggesting a similar compound could be bound)?

In the revised manuscript we have included a new Figure S12, which illustrates the conservation of sequence in the Ex1 and Ex2 cavities (analysis performed using the ConSurf server with a multiple sequence alignment of 180 mycobacterial homologues of Cya). The corresponding part of the revised text starts with line 235:

“Interestingly, only two of the site Ex1 residues are well conserved, based on the alignment of 180 close homologues of Cya (Figure S8, S12A): while T122 and Q127 are relatively well conserved, the residues D123, E164 and D170 are not. Although our evidence points to a possibility of *M. tuberculosis* Cya Ex1 site’s involvement in cation binding, this region may play distinct roles in the Cya homologues of other mycobacterial species. Likewise, the site Ex2 is lined by largely nonconserved residues (Figure S12B). Developing a more precise understanding of this site’s functional role will be a prerequisite for understanding the significance of the few conserved residues in this pocket (F87, A109, G125 in the *M. tuberculosis* Cya).”

3) The authors suggest site Ex1 binds a metal and that this may play a sensing role. if the authors make this claim I think they should provide evidence that the occupancy of the site can change under the expected physiological concentration changes. Can MD provide some hints to the metal affinity? Can changing buffers or adding chelating agents remove the ion?

This is an extremely challenging problem to address experimentally at the moment, because to determine the occupancy changes at the extracellular site we would need to isolate the intracellular site. We are very keen to do this using, for example, liposome-reconstituted purified Cya – subjecting the samples to single particle cryo-EM or cryo-ET and subtomogram averaging. These approaches will be the focus of the future studies in our laboratory. Due to the complexity and technical challenges these experiments fall outside the scope of the current study.

However, to characterize the properties of the Ex1 site, we have performed MD simulations with the Ex15A mutant. The results are shown in the new Figure S13. Mutation of the site abolishes cation binding, without perturbing the canonical metal binding site in the ATP binding pocket of the protein. The description and our interpretation of the observations is on line 243:

“To further support our insights derived from the experiments with the Ex1-5A mutant we performed the MD simulations with Ex1-5A protein (Figure S13), using the identical protocol as those used for the wild-type Cya (Figure S10). The simulation revealed a dramatic difference between the B-factors of the Ex1-5A and the wild-type protein, indicative of increased conformational flexibility of the mutant (Figure S13A). As expected, metal ion binding was preserved at the cytosolic domain of the Ex1-5A, but was completely disrupted at the extracellular Ex1 site (Figure S13B-C). Analysis of the distances between select residues in Cya and in Ex15A showed that although the core remains relatively stable for both protein, the TM1-5 region may be more conformationally flexible (judged by the increased average distances between the residues A77-A77 and L92-L92 within the dimer; Figure S13D-E). The same is true for portions of the catalytic domain, evident from the increased average distance in the residue Q289. Together with the functional data, the results of the MD simulations strongly suggest that disruption of the extracellular surface of Cya (site Ex1) leads to profound changes within the intramembrane part of the protein, as well as at a distant catalytic domain.”

Is there any evidence for extracellular ion sensors?

The extracellular ion sensors indeed exist. In mammals a prominent example is the calcium sensing receptor (CaSR), a G-protein coupled receptor. In bacteria, salt sensing has been shown in histidine kinases (*Sphingomonas melonis* KipF, Kaczmarczyk et al., 2015; doi: 10.1128/JB.00019-15) or in chemotaxis receptors (*Escherichia coli* Tar, Shuangyu et al., 2018; doi: 10.1038/s41467-018-05335-w). We have added this as a point in the discussion (line 289):

“It is worth noting that sensing of extracellular metals is a well established in eukaryotic and prokaryotic cell signaling. For example, a prominent example in mammalian signaling pathway is the calcium-sensing receptor (CaSR), a GPCR involved in regulation of Ca2+ homeostasis (29). In bacteria, salt sensing has been shown in histidine kinases (Sphingomonas melonis KipF) (30) and in chemotaxis receptors (*Escherichia coli* Tar) (31). Thus, a potential link between extracellular metal ion sensing and cAMP signaling that may be mediated by the M. tuberculosis Cya is within the realm of possibility and presents an interesting avenue for future investigations.”